# Classical boundary field theory of Jacobi sigma models by Poissonization

**Ion V. Vancea**

*Group of Theoretical Physics and Mathematical Physics, Department of Physics*
*Federal Rural University of Rio de Janeiro*
*Cx. Postal 23851, BR 465 Km 7, 23890-000 Seropédica - RJ, Brazil*
`ionvancea@ufrrj.br`

9 April 2021

## Abstract

In this paper, we are going to construct the classical field theory on the boundary of the embedding of $\mathbb{R} \times S^1$ into the manifold $M$ by the Jacobi sigma model. By applying the poissonization procedure and by generalizing the known method for Poisson sigma models, we express the fields of the model as perturbative expansions in terms of the reduced phase space of the boundary. We calculate these fields up to the second order and illustrate the procedure for contact manifolds.

**Keywords**: Jacobi sigma model; Poisson sigma model; boundary field theory.

Prepared for the Proceedings of *4th International Conference on Holography, String Theory and Discrete Approach*, Hanoi, Vietnam, 2020.

# 1 Introduction

Very recently, the Jacobi sigma models have been realized as embeddings into the Poisson sigma models in a target space of one dimension higher [1, 2] by using a procedure called *poissonization* [3]. The poissonization method has been used to study several important problems of Jacobi structures of interest in mathematics and physics such as the reduction of the homogeneous tensors [4], the integrability [5], the supergeometry formulation [6], the deformation of their coisotropic submanifolds [7] and the contact structures with singularities [8].

In this paper, we are going to apply the poissonization technique to obtain the classical field theory on the boundary $\partial\Sigma$ of the base space $\Sigma = S^1 \times [0, L]$ of a Jacobi sigma model with the target space $M$. The interest in this type of field theories is on their interpretation as holographic dual fields to the Jacobi sigma models upon quantization. By extension, we call the poissonization of the Jacobi sigma model $z : \Sigma \to M$ a Poisson sigma model $z : \Sigma \to M_+$, where $M_+ = \mathbb{R} \times M$. In local coordinates, the fields of the Poisson sigma model are $z^I = (z^0, z^i)$ with $z^i$ denoting the fields of the Jacobi sigma model, $i = 1, 2, \ldots, \dim(M)$ and $M$ being a Jacobi manifold. The Jacobi structure on a differentiable manifold was introduced by Lichnerowicz in [3] as a generalization of the Poisson manifold and a local realization of the Kirillov algebras [9] (see also [5]). In general, the poissonization procedure refers to the embedding of a Jacobi manifold into a higher dimensional Poisson manifold. Here, we are going to use this embedding to derive a lower dimensional field theory from the classical field theory of the Poisson sigma model associated to the Jacobi sigma model. The field theory on $\partial\Sigma$ for the Poisson sigma model with $\Sigma = S^1 \times [0, L]$ was discussed previously in [10, 11]. The same problem on the disk was addressed in [12].

The paper is organized as follows. In Section 2, we are going to review the Jacobi structure and the poissonization procedure and outline some of their properties. In Section 3, we are going to present the construction of the Jacobi sigma model by poissonization following the recent results from [1, 2]. In Section 4, we will derive the classical boundary field theories associated to the Jacobi sigma models by poissonization of the classical field theory of the corresponding higher dimensional Poisson sigma model. Here, we obtain the classical action and calculate the classical fields as a power expansion with coefficients that depend on the the reduced boundary phase space. Also, we give the explicit formulas for the fields up to the second order in the expansion and illustrate the results in the case of the contact manifolds. Our discussion follows closely the analysis from [11] of the Poisson sigma model. In the last section we conclude the paper.

# 2  Jacobi structures

The Jacobi manifolds were introduced in [3] as a natural generalization of the Poisson manifolds. In this section, we are going to review some basic definitions, properties and examples of the Jacobi structures following mainly [13, 14].

**Definition 1.** A *Jacobi structure* is the triplet $(M, \pi, R)$ where $M$ is a differentiable manifold, $\pi$ is a bivector from $\Gamma(\Lambda^2(T(M)))$ and $R$ is a vector field from $\Gamma(T(M))$ such that

$$[\pi, \pi] = 2R \wedge \pi \quad \text{and} \quad \mathcal{L}_R \pi = [R, \pi] = 0 \,. \tag{1}$$

Here, $[\cdot, \cdot]$ is the Schouten-Nijenhuis bracket $[\cdot, \cdot] : \Lambda^p(M) \times \Lambda^q(M) \to \Lambda^{p+q-1}(M)$ and $R$ is the Reeb vector field. The manifold $M$ with the Jacobi structure is called a *Jacobi manifold*. A *Jacobi bracket* can be defined on the set $C^\infty(M)$ by the following relation

$$\{f, g\} = \pi(df, dg) + fR(g) - gR(f)\,, \quad \forall f, g \in C^\infty(M)\,. \tag{2}$$

The Jacobi brackets satisfy the following relations

$$\{c_1 f + c_2 g, h\} = c_1 \{f, g\} + c_2 \{f, g\}\,, \tag{3}$$

$$\{f, g\} = -\{g, f\}\,, \tag{4}$$

$$\{\{f, g\}, h\} + \{\{g, h\}, f\} + \{\{h, f\}, g\} = 0\,, \tag{5}$$

$$\{f, gh\} = g\{f, h\} + h\{f, g\} + gh\,(Rf)\,, \tag{6}$$

for all $f, g, h \in C^\infty(M)$ and all $c_1, c_2 \in \mathbb{R}$. Since

$$\operatorname{supp}\{f, g\} \subseteq \operatorname{supp} f \cap \operatorname{supp} g\,, \tag{7}$$

for all $f, g \in C^\infty(M)$, it follows that $(C^\infty(M), \{\cdot, \cdot\})$ is a local Lie algebra [9]. We note that the Jacobi structure induces the following bundle map

$$(\pi, R)^\sharp : T^\star(M) \times \mathbb{R} \to T(M) \times \mathbb{R}\,, \tag{8}$$

$$(\pi, R)^\sharp (\omega, v) = \left( \pi^\sharp \omega + vR, -\langle \omega, R \rangle \right)\,, \tag{9}$$

for all $\omega, v \in \Gamma\left(T^\star(M) \times \mathbb{R}\right)$ where $v$ acts multiplicatively on $R$.

**Examples.** Well know examples of Jacobi structures are the symplectic Poisson structures, the vectors fields on manifolds for which the bivector vanishes, the locally conformal symplectic manifolds and the contact structures. Let us see some of these examples in more detail.

1. *Locally conformal symplectic manifolds.* The locally conformal sympletic structure is defined on an even dimensional symplectic manifold $M$ of dimension $2m$ endowed with a non-degenerate 2-form $\beta \in \Lambda^2(M)$ and an 1-form $\gamma \in \Lambda^1(M)$ that satisfy the following equations

$$d\beta + \beta \wedge \gamma = 0 \,, \tag{10}$$
$$d\gamma = 0 \,. \tag{11}$$

   The other two components of the triplet $\pi$ and $R$ are defined by the following relations

$$i_R\beta = \gamma \,. \tag{12}$$
$$i_{\pi\omega}\beta = -\omega \,, \tag{13}$$

   for all $\omega \in \Lambda^1(M)$. For any point $x \in M$, there is an open set $U_x \subset M$ and a function $f : U_x \to \mathbb{R}$ such that $\gamma = df$ and $e^f\beta$ is symplectic.

2. *Contact manifolds.* The contact manifolds and their Jacobi structure are defined as follows [15]. Let $M$ be an odd dimensional manifold of dimension $(2m + 1)$ and $\gamma \in \Lambda^1(M)$. Then $\gamma$ is a contact form if it satisfies the following relation

$$\gamma \wedge (d\gamma)^m \neq 0 \,, \tag{14}$$

   at every point of $M$. The pair $(M, \gamma)$ is a contact manifold.

   Let $\flat$ be the following isomorphism of $C^\infty(M, \mathbb{R})$-modules

$$\flat : T(M) \longrightarrow \Lambda^1(M) \,, \tag{15}$$
$$\flat(X) = i_X d\gamma + \gamma(X)\,\gamma \tag{16}$$

   Then the bivector $\pi$ and the Reeb vector $R$ are defined by the following relations

$$\pi(\omega, \eta) = d\gamma\left(\flat^{-1}(\omega), \flat^{-1}(\eta)\right) \,, \tag{17}$$
$$R = \flat^{-1}(\gamma) \,, \tag{18}$$

   for all $\omega, \eta \in \Lambda^1(M)$. In particular, the Reeb vector field satisfies the following equations

$$i_R\gamma = 1 \,, \quad i_R d\gamma = 0 \,. \tag{19}$$

In general, the Jacobi structure reduces to a Poisson structure for a vanishing Reeb vector field $R = 0$. This defines a rather trivial relation between the Jacobi and the Poissson

structures on a given manifold. However, there is a non-trivial connection between the Jacobi structure and the Poisson structure given by the poissonization.

**Definition 2.** The *poissonization* of the Jacobi structure $(M, \pi, R)$ is the Poisson structure $(M_+, \alpha)$ where $M_+ = \mathbb{R} \times M$ and the Poisson bivector $\alpha$ is defined as follows

$$\alpha = e^{-t} (\pi + \partial_t \wedge R) , \tag{20}$$

where $t$ is the canonical coordinate on $\mathbb{R}$.

According to the above definition, the poissonization of the Jacobi structure $(M, \pi, R)$ is the structure $(M_+, \alpha, \partial_t)$. The Hamiltonian vector fields play an important role in the Jacobi as well as in the Poisson structures. Therefore, let us give their definition below.

**Definition 3.** Let $(M, \pi, R)$ be a Jacobi structure and $f \in C^\infty(M, \mathbb{R})$. Then the *Hamiltonian vector field* associated to $f$ is defined as

$$X_f = \pi(df) + fR . \tag{21}$$

The Hamiltonian vector fields on the Jacobi structure $(M, \pi, R)$ define the *characteristic distribuition* of $M$ as follows

$$\mathsf{C}(M) := \mathrm{Span} \left\{ R, \pi\omega \,|\, \omega \in \Lambda^1(M) \right\} \subseteq T(M) . \tag{22}$$

If $\mathsf{C}(M) = T(M)$, the Jacobi structure is said to be transitive. In general, $\mathsf{C}(M)$ is integrable and defines a foliation on $M$.

**Definition 4.** Let $(A, \rho, M)$ be a vector bundle over $M$. Then $A$ is said to be a *Lie algebroid* if there is a Lie bracket $[\cdot, \cdot]_A$ on the space of smooth sections $\Gamma(A)$ and a bundle map $\rho : A \to T(M)$ such that

$$\rho \left([X, Y]_A\right) = [\rho(X), \rho(Y)] , \tag{23}$$

$$[X, fY]_A = f [X, Y]_A + (\rho(X)f) (Y) , \tag{24}$$

for any $X, Y \in \Gamma(A)$ and any $f \in C^\infty(M)$. The map $\rho$ is called the *anchor* of the algebroid $A$.

One can associate a Lie algebroid to any Jacobi structure by considering that $A = T^\star(M) \oplus \mathbb{R}$ and defining the following Lie bracket on $\Gamma(T^\star(M))$

$$\{(\omega, f), (\gamma, g)\}_{(\pi, R)} = (\mathcal{L}_{\pi\omega}\gamma - \mathcal{L}_{\pi\gamma}\omega - d\left(\pi(\omega, \gamma)\right) + f\mathcal{L}_R\gamma - g\mathcal{L}_R\omega - i_R \left(\omega \wedge \gamma\right) ,$$

$$\pi(\omega, \gamma) + \pi(\omega, dg) - \pi(\gamma, df) + fR(dg) - gR(df)) \, . \tag{25}$$

The anchor map of the Jacobi algebroid is given by the following relation

$$\rho(\omega, f) = \pi\omega = fR \, . \tag{26}$$

For more details on these constructions see, e. g. [13, 14, 16].

# 3 Jacobi sigma models

In this section, we are going to review the construction of the Jacobi sigma models by poissonization as given in [1, 2].

## 3.1 Poisson sigma models

Consider a Jacobi structure $(M, \pi, R)$. This structure can be lifted by poissonization to the Poisson structure $(M_+, \alpha)$ as discussed in the previous section. Without loss of generality, we can choose the embedding $M = \{0\} \times M \in M_+$ of $M$ at the origin of the factor $\mathbb{R}$.

Let us give a formal definition of the Poisson sigma models [17, 18].

**Definition 5.** The *Poisson sigma model* is defined by the following set of objects:

1. A *base space* $\Sigma$ which is a two-dimensional smooth manifold of boundary $\partial\Sigma$.

2. A *target space* $M$ which is a finite dimensional smooth manifold endowed with a Poisson structure $(M, \alpha)$.

3. A *space of fields* $\Phi$ of the model which is the space of morphisms of vector bundles $\mathrm{Mor}^k\left(T(\Sigma), T^\star(M)\right)$ and are continuous of class $C^k$.

4. A *local action functional* constructed as follows. Consider a $(z, \eta)$ parametrization of $\mathrm{Mor}^k\left(T(\Sigma), T^\star(M)\right)$ where $z \in \mathrm{Mor}^{k+1}\left(\Sigma, M\right)$ define the embedding of the base space into the target space and $\eta \in \Gamma^k\left(\Sigma, T^\star(\Sigma) \times z^\star T^\star(M)\right)$ denote the space of differentiable sections of class $C^k$. Then the first order action of the Poisson sigma model is given by the following functional

$$S[z, \eta] = \int_\Sigma \langle \eta, dz \rangle + \frac{1}{2}\left\langle \eta, \left(\alpha^\sharp \circ z\right)\eta \right\rangle \, . \tag{27}$$

Here, we are using the following notations. The bracket $\langle \cdot, \cdot \rangle$ denotes the pairing of elements from $\Lambda^1\left(\Sigma, z^\star T^\star(M)\right)$ and $\Lambda^1\left(\Sigma, z^\star T(M)\right)$, $\circ$ denotes the composition of maps

and

$$\alpha^\sharp : T^\star(M) \to T(M), \quad \omega \mapsto \alpha(\omega, \cdot). \tag{28}$$

As mentioned above, we consider that $\eta \in \Lambda^1\left(\Sigma, z^\star T^\star(M)\right)$ and $dz \in \Lambda^1\left(\Sigma, z^\star T(M)\right)$.

Note that if the manifold $\Sigma$ has a boundary $\partial\Sigma$, then the space of fields $\Phi|_{\partial\Sigma}$ contains the morphisms of class $C^k$ between $T(\partial\Sigma)$ and $T^\star(M)$. The set of these morphisms form a fiber bundle $P(T^\star(M))$ with fibers $T_z^\star(P(M))$ over the space of paths $P(M)$ in $M$. The identification between $T^\star(M)$ with $P(T^\star(M))$ is given by the following map

$$\phi : T^\star(P(M)) \to P(T^\star(M)), \quad (z, \eta) \mapsto c(\tau) = (z(\tau), \eta(\tau)), \tag{29}$$

where $c(\tau)$ is a path. This space can also be endowed with a symplectic form $\omega_\partial$ such that for any curve $c$

$$\omega_\partial[c](X_1, X_2) = \int_0^1 d\tau \, \omega_0(X_1(\tau), X_2(\tau)) , \tag{30}$$

where $\omega_0$ is the canonical symplectic form on $T^\star(M)$ and $X_1$ and $X_2$ are tangent vectors to the curve $c(\tau)$.

For the calculations in field theory, it is convenient to introduce local coordinates on both base and target spaces, respectively. In what follows, we are going to use $\sigma^a = (\sigma^0, \sigma^1)$ as local coordinates on $\Sigma$ and $z^i = (z^1, \ldots, z^m)$ as local coordinates on $M$. By applying the variational principle to the action (27), we obtain the following equations of motion

$$dz^i + \alpha^{ij}(z)\eta_j = 0 , \tag{31}$$

$$d\eta_i + \frac{1}{2}\partial_i \alpha^{jk}(z)\eta_j \wedge \eta_k = 0 . \tag{32}$$

The above equations represent the condition for the vector bundle morphisms to be the morphisms of the corresponding Lie algebroids [19] [1].

The action (27) is invariant up to a total derivative term under the infinitesimal gauge transformations given by the following relations

$$\delta_\epsilon z^i = \alpha^{ij}(z)\epsilon_j , \tag{33}$$

$$\delta_\epsilon \eta_i = -d\epsilon_i - \partial_i \alpha^{jk}(z)\eta_j \epsilon_k , \tag{34}$$

where $\epsilon \in \Gamma^k\left(z^\star T^\star(M)\right)$ is the gauge parameter. The total derivative term has the following form

$$\delta_\epsilon S[z, \eta] = -\int_\Sigma d\left(dz^i \epsilon_i\right) . \tag{35}$$

---

[1] I acknowledge T. Strobl for pointing this out to me

The variation $\delta_\epsilon S[z, \eta] = 0$ if $\partial\Sigma = \emptyset$. For the particular choice $\epsilon = \langle\zeta, \eta^\sharp\rangle_\Sigma$, where the contraction is with respect to the cotangent and tangent spaces of the base manifold and $\zeta \in \Gamma^k(T(\Sigma))$, the gauge transformations (33) and (34) take the following form

$$\delta_\epsilon z^i = \mathcal{L}_\zeta z^i - i_\zeta \left(dz^i + \alpha^{ij}(z)\epsilon_j\right), \tag{36}$$

$$\delta_\epsilon \eta_i = \mathcal{L}_\zeta \eta_i - i_\zeta \left(d\eta_i + \frac{1}{2}\partial_i\alpha^{jk}(z)\eta_j \wedge \eta_k\right). \tag{37}$$

From the equations (36) and (37), we can concluded that the action $S[z, \eta]$ is invariant under the local diffeomorphisms on-shell. However, the gauge symmetries are not covariant with respect to the target space transformation of the coordinates. In order to remedy this, an torsion-free connection term can be introduced [20].

## 3.2   Jacobi sigma models

The Jacobi sigma model with the target space given by the Jacobi structure $(M, \pi, R)$ was defined in [1, 2] such that its dynamics coincide with the dynamics of the poissonization sigma model (in the sense defined in the previous section) with the target space given by the Poisson structure $(M_+, \alpha)$ at $t = 0$. In this definition, $\alpha$ is given by the relation (20). These requirements are satisfied by the following action functional

$$S_J[z, \eta] = \int_\Sigma \langle\eta, dz\rangle + \frac{1}{2}\langle\eta, (\pi \circ z)\,\eta\rangle + \langle(R \circ z)\,\eta, \lambda\rangle, \tag{38}$$

where for simplicity we have identified notationally the sharp quantities with the corresponding un-sharp ones. Here, $\lambda \in \Gamma^k\left(\Sigma, T^\star(\Sigma) \times z^\star T^\star(M)\right)$ is an auxiliary field introduced by the poissonization procedure.

We use the coordinate system introduced in the previous subsection with the coordinate $z^0$ on $\mathbb{R}$. Then $M_+ = \mathbb{R} \times M$ has the local coordinates $z^I = (z^0, z^i)$. One can easily see that the action (38) takes the following form in these coordinates

$$S_J[z, \eta] = \int_\Sigma d^2\sigma\, \varepsilon^{ab}\left(\eta_{a0}\partial_b z^0 + \eta_{ai}\partial_b z^i + \frac{1}{2}e^{-z^0}\pi^{ij}(z)\eta_{ai}\eta_{bj} + e^{-z^0}R^i(z)\eta_{a0}\eta_{bi}\right). \tag{39}$$

As usual, one can derive the equations of motion from the action (39) by applying the variational principle. The result is the following set of equations

$$\partial_a z^0 + e^{-z^0}R^i\eta_{ai} = 0, \tag{40}$$

$$\partial_a z^i + e^{-z^0}\pi^{ij}\eta_{aj} - e^{-z^0}R^i\eta_{a0} = 0, \tag{41}$$

$$\partial_a\eta_{b0} - \frac{1}{2}e^{-z^0}\pi^{ij}\eta_{ai}\eta_{bj} - e^{-z^0}R^i\eta_{a0}\eta_{bi} = 0, \tag{42}$$

$$\partial_a \eta_{bi} + \frac{1}{2} e^{-z^0} \partial_i \pi^{jk} \eta_{aj} \eta_{bk} + e^{-z^0} \partial_i R^j \eta_{a0} \eta_{bj} = 0 \,. \tag{43}$$

As was noted in [2], if the defining conditions of the Jacobi structure given by the relations (1) are satisfied, then the action $S_J[z, \eta]$ is invariant under the following infinitesimal gauge transformations

$$\delta_\epsilon z^0 = -e^{-z^0} R^i \epsilon_i \,, \tag{44}$$

$$\delta_\epsilon z^i = e^{-z^0} \left( -\pi^{ij} \epsilon_j + R^i \epsilon_0 \right) \,, \tag{45}$$

$$\delta_\epsilon \eta_{a0} = \partial_a \epsilon_0 - e^{-z^0} \pi^{jk} \eta_{aj} \epsilon_k + e^{-z^0} R^j \left( \eta_{aj} \epsilon_0 - \eta_{a0} \epsilon_j \right) \,, \tag{46}$$

$$\delta_\epsilon \eta_{ai} = \partial_a \epsilon_i + e^{-z^0} \partial_i \pi^{jk} \eta_{aj} \epsilon_k - e^{-z^0} \partial_i R^j \left( \eta_{aj} \epsilon_0 - \eta_{a0} \epsilon_j \right) \,, \tag{47}$$

where the gauge parameters $\epsilon_0$ and $\epsilon_i$ are two-dimensional smooth scalar functions. Also, the algebra of these transformations closes only on-shell

$$[\delta_{\epsilon^1}, \delta_{\epsilon^2}] z^0 = \delta_{\epsilon^3} z^0 \,, \tag{48}$$

$$[\delta_{\epsilon^1}, \delta_{\epsilon^2}] z^i = \delta_{\epsilon^3} z^i \,, \tag{49}$$

$$[\delta_{\epsilon^1}, \delta_{\epsilon^2}] \eta_{a0} = \delta_{\epsilon^3} \eta_{a0} - \epsilon_I^3 \left( \text{eq. motion } z^I \right) \,, \tag{50}$$

$$[\delta_{\epsilon^1}, \delta_{\epsilon^2}] \eta_{ai} = \delta_{\epsilon^3} \eta_{ai} - \epsilon_i^3 \left( \text{eq. motion } z^0 \right)$$
$$+ e^{-z^0} \left( \partial_i \partial_j \pi^{kl} \epsilon_k^1 \epsilon_l^2 - \partial_i \partial_j R^k \left( \epsilon_k^1 \epsilon_0^2 - \epsilon_k^2 \epsilon_0^1 \right) \right) \left( \text{eq. motion } z^i \right) \,. \tag{51}$$

The general formulation of the Jacobi sigma model given above as a poissonization model considers the target space $M_+$. In order to define the Jacobi sigma model on $M$, one has to consider the embedding $M \in M_+$ at $z^0 = 0$ [1] which introduces a certain simplification in the general case. Also, in order to cancel the gauge variation of $z^0$ from the gauge algebra, one should require that the gauge transformations of $z^0$ vanish. The same considerations apply to the conjugate field variable $\eta_0$. These requirements impose the following constraints on the remaining fields

$$R^i \eta_{ai} = 0 \,, \tag{52}$$

$$\pi^{ij} \eta_{ai} \eta_{bj} = 0 \,, \tag{53}$$

$$R^i \epsilon_i = 0 \,, \tag{54}$$

$$\partial_a \epsilon_0 - \pi^{jk} \eta_{aj} \epsilon_k + R^j \eta_{aj} \epsilon_0 = 0 \,. \tag{55}$$

The above construction shows that the Jacobi sigma model has a simple form after the poissonization procedure is applied. Also, we can use the correspondence between the Jacobi and the Poisson sigma models to derive some properties of the classical Jacobi field theory from

the corresponding properties of the Poisson field theory.

# 4 Classical boundary field theory of Jacobi sigma model

In this section, we are going to derive the classical boundary field theory of the Jacobi sigma model by applying the poissonization procedure discussed above. Since the field theory lives on the boundary, the choice of the base manifold $\Sigma$ determines its general properties.

The starting point is the Jacobi sigma model on cylinder $\Sigma = S^1 \times [0, L]$ defined by the following action

$$S[z, \eta] = \int_\Sigma d^2\sigma \varepsilon^{ab} \left( z^0 \partial_a \eta_{b0} + z^i \partial_a \eta_{bi} + \frac{1}{2} e^{-z^0} \pi^{ij}(z) \eta_{ai} \eta_{bj} + e^{-z^0} R^i \eta_{a0} \eta_{bi} \right) . \tag{56}$$

The action has the full target space $M_+$ with $z$ denoting the coordinates $z^i$ on $M$ submanifold. The equations of motion, the gauge symmetries and the gauge algebra were given in the Section 3.2 above (with an overall minus sign of the gauge parameter).

In order to find the classical field theory of the Jacobi sigma model on the boundary, we apply the poissonization method to extract the relevant information from the field theory localized on the boundary of the corresponding Poisson sigma model. We follow the general procedure of constructing solutions of the classical equations of motion with given boundary conditions given in [10, 11].

The first step is to perform a gauge fixing of the Jacobi sigma model on $M$ such that the gauge be consistent with the gauge chosen for the Poisson sigma model on $M_+$. This condition ensures that the field theory on the boundary obtained from the Jacobi sigma model is mapped into the field theory on the boundary of its poissonization. The latter has been analyzed in [11]. The correspondence between the two field theories requires picking the boundary values of the fields $\eta_{a0}$ and $\eta_{ai}$ and of the parameters $\epsilon_0$ and $\epsilon_i$ such that the gauge variation of the action and the gauge variations of $\eta_{a0}$ and $\eta_{ai}$ vanish on the boundary.

The second step is to analyse the variation of the action under the gauge transformations on the boundary $\partial\Sigma = S^1 \times \{0, L\}$ which has the following form

$$\delta_\epsilon S[z, \eta] = \int_0^{2\pi} d\sigma^0 \left\{ e^{-z^0} \left[ \left( R^i + z^0 R^i - z^j \partial_j R^i \right) \left( \eta_{00} \varepsilon_i - \eta_{0i} \varepsilon_0 \right) \right. \right.$$
$$\left. \left. + \eta_{0i} \varepsilon_j \left( \pi^{ij} + z^0 \pi^{ij} - z^k \partial_k \pi^{ij} \right) \right] \right\} \Big|_0^L . \tag{57}$$

The variation of the action $\delta_\epsilon S[z, \eta]$ and the variation of the fields $\eta_{a0}$ and $\eta_{ai}$ vanish if the

following equations are satisfied

$$\eta_{00}|_{\partial\Sigma} = \eta_{0i}|_{\partial\Sigma} = 0\,, \quad \epsilon_0|_{\partial\Sigma} = \epsilon_i|_{\partial\Sigma} = 0\,. \tag{58}$$

In the next step, we consider a classical background that is close to the trivial solutions of the equations of motion as in [11, 12]

$$z^0 = z^i = 0\,, \quad \eta_{a0} = \eta_{ai} = 0\,. \tag{59}$$

The variations of the fields in this background read

$$\delta_\epsilon z^0 = R^i \epsilon_i\,, \quad \delta_\epsilon z^i = \pi^{ij}\epsilon_j - R^i\epsilon_0\,, \tag{60}$$

$$\delta_\epsilon \eta_{a0} = -\partial_a\epsilon_0\,, \quad \delta_\epsilon \eta_{ai} = -\partial_a\epsilon_i\,. \tag{61}$$

From the equations (58) and (61) we conclude that

$$\eta_{10}(\sigma) = \phi_0(\sigma^0)\,, \quad \eta_{1i}(\sigma) = \phi_i(\sigma^0)\,. \tag{62}$$

By correspondence to the Poisson sigma model, the generators of the gauge transformation are the constraints

$$\frac{\partial z^0}{\partial \sigma^1} + e^{-z^0} R^i \phi_i = 0\,, \tag{63}$$

$$\frac{\partial z^i}{\partial \sigma^1} - e^{-z^0} R^i \phi_0 + e^{-z^0}\pi^{ij}\phi_j = 0\,. \tag{64}$$

Since the action $S[z,\eta]$ is presented in the first order formalism, we can employ the Fadeev-Jackiw method to determine the reduced phase space variables [11]. To this end, we consider the following initial conditions that could select a single solution from each gauge orbit

$$z^0(\sigma^0, \sigma^1 = 0) = \zeta^0(\sigma^0)\,, \qquad z^i(\sigma^0, \sigma^1 = 0) = \zeta^i(\sigma^0)\,. \tag{65}$$

The general form of the solutions in the vicinity of the initial conditions is

$$z^0\left(\sigma^1; \phi_0(\sigma^0), \phi_i(\sigma^0), \zeta^0(\sigma^0), \zeta^i(\sigma^0)\right) = z^0\left(\sigma^1; \phi_I, \zeta^I\right)\,, \tag{66}$$

$$z^i\left(\sigma^1; \phi_0(\sigma^0), \phi_i(\sigma^0), \zeta^0(\sigma^0), \zeta^i(\sigma^0)\right) = z^i\left(\sigma^1; \phi_I, \zeta^I\right)\,. \tag{67}$$

By using $z^0$ and $z^i$ from the relations (66) and (67) we can define the following boundary

fields

$$x^0(\sigma^0) = x^0(\phi_I, \zeta^I) \equiv \int_0^L d\sigma^1 \, z^0\left(\sigma^1; \phi_I, \zeta^I\right), \tag{68}$$

$$x^i(\sigma^0) = x^i(\phi_I, \zeta^I) \equiv \int_0^L d\sigma^1 \, z^i\left(\sigma^1; \phi_I, \zeta^I\right). \tag{69}$$

Note that the fields $x^0$ and $x^i$ are implicit functions on $\sigma^0$ through the reduced phase space variables $(\phi_I, \zeta^I)$.

The above data allows one to determine the action of the boundary field theory which is given by the action (56) with the boundary conditions (62) and depends on the fields $x^0$ and $x^i$. It is easy to see that the sought for action has the following form

$$S_{\text{bound}}[x, \phi] = -\int_{S^1} d\sigma^0 \left[x^0 \frac{d\phi_0}{d\sigma^0} + x^i \frac{d\phi_i}{d\sigma^0}\right]. \tag{70}$$

The action $S_{\text{bound}}[x, \phi]$ describes an one-dimensional classical field theory on the boundary of $\Sigma = S^1 \times [0, L]$. Since this is the dual field to the poissonization of the Jacobi sigma model, we call it by extension the poissonization of the dual field to the Jacobi sigma model.

The action $S_{\text{bound}}[x, \phi]$ has the same form as the classical action of the Poisson sigma model obtained in [11] which is no coincidence since this is the poissonization of the Jacobi model we started with. The dependence of the Lagrangian from (70) on the Jacobi structure is implicit in the construction of the canonically dual variables $\{x^I, \phi_I\}$. As shown above, these variables are solutions to the equations (63) and (64) which are more difficult to solve than in the Poisson case since they are non-linear in general. However, one can always search for an approximate solution if a formal power expansion is defined as in [11]. To this end, one introduces a real positive parameter $\lambda$ which will be set to one at the end of the calculations. The modified equations are

$$\frac{\partial z^0}{\partial \sigma^1} + \lambda e^{-z^0} R^i(z)\phi_i = 0, \tag{71}$$

$$\frac{\partial z^i}{\partial \sigma^1} - \lambda e^{-z^0} R^i(z)\phi_0 + \lambda e^{-z^0} \pi^{ij}(z)\phi_j = 0. \tag{72}$$

The fields $z^0$ and $z^i$ are assumed to depend on $\lambda$. Then the formal series in powers of $\lambda$ have the following form

$$z^0 = x^0 + \sum_{n=1}^{\infty} \lambda^n z_n^0, \tag{73}$$

$$z^i = x^i + \sum_{n=1}^{\infty} \lambda^n z_n^i, \tag{74}$$

$$\pi^{ij}(z) = \pi^{ij}(x) + \sum_{n=1}^{\infty} \lambda^n \frac{\partial^{n_1+n_2+\cdots n_s}}{\partial^{n_1} z^{i_1} \partial^{n_2} z^{i_2} \cdots \partial^{n_s} z^{i_s}} \pi^{ij}(z)\big|_{z=x} z^{i_1}_{n_1} z^{i_2}_{n_2} \cdots z^{i_s}_{n_s}, \qquad (75)$$

$$R^i(z) = R^i(x) + \sum_{n=1}^{\infty} \lambda^n \frac{\partial^{n_1+n_2+\cdots n_s}}{\partial^{n_1} z^{i_1} \partial^{n_2} z^{i_2} \cdots \partial^{n_s} z^{i_s}} R^i(z)\big|_{z=x} z^{i_1}_{n_1} z^{i_2}_{n_2} \cdots z^{i_s}_{n_s}, \qquad (76)$$

with $n_1 + n_2 + \cdots + n_s = n$ and in the equations (75 ) and (76) there is a sum over each index $i_k$. The initial conditions are defined as follows

$$z^0_0 = x^0, \quad z^0_n = 0, \qquad (77)$$

$$z^i_0 = x^i, \quad z^i_n = 0, \quad n \in \mathbb{N}. \qquad (78)$$

For an arbitrary $z^0$, the expansion of the exponential factor spoils the power expansion of the modified constraints. Therefore, the method is valid strictly at the point $z^0 = 0$ where the Jacobi structure is embedded into the poissonization. Then a simple algebra gives the constraints and solutions at all orders in $\lambda$.

Let us write down the corresponding relations at the first orders. At order zero, there is no constraint and the solution is the initial condition

$$z^i_0 = x^i(\sigma^0). \qquad (79)$$

At first order, the constraint and the solution are

$$R^i(x)\phi_i = 0, \qquad (80)$$

$$z^i_1 = \left[ R^i(x)\phi_0 - \pi^{ij}(x)\phi_j \right] \sigma^1. \qquad (81)$$

At second order, the constraint and the solutions have the following form

$$\frac{\partial R^i(x)}{\partial z^j}\phi_i = 0, \qquad (82)$$

$$z^i_2 = \left[ \frac{\partial R^i(x)}{\partial z^j} R^j(x) (\phi_0)^2 - \left( \frac{\partial R^i(x)}{\partial z^j}\pi^{jk}(x) + \frac{\partial \pi^{ik}(x)}{\partial z^j}R^j(x) \right) \phi_0 \phi_k \right.$$
$$\left. + \frac{\partial \pi^{ij}(x)}{\partial z^k}\pi^{kr}(x)\phi_r\phi_j \right] \left( \sigma^1 \right)^2. \qquad (83)$$

Similarly, we can solve the equations at arbitrary higher order in $\lambda$ which determines the classical boundary field theory completely.

As an example, consider the contact manifold $(M,\gamma)$ with $\dim M = 2m+1$. As discussed in Section 2, there is an induced Jacobi structure $(M,\pi,R)$. The poissonization of the contact manifold has the dimension $\dim M_+ = 2m+2$. We denote by $z^i = (t, z^r, z^{m+r})$, $r, s = 1, \ldots, m$

the canonical coordinates of $(M, \pi, R)$. Then one can see that at order $0, 1, 2$ in $\lambda$ the relations (79) - (83) take the following form

$$t_0 = t(\sigma^0)\,, \qquad\qquad z_0^r = x^r(\sigma^0)\,, \quad z_0^{m+r} = x^{m+r}(\sigma^0)\,, \qquad (84)$$

$$t_1 = \left[\phi_0 - x^{m+s}\phi_{m+s}\right]\sigma^1\,, \qquad z_1^r = \phi_r\sigma^1\,, \quad z_1^{m+r} = \left[-x^{m+r}\phi_t + \phi_r\right]\sigma^1\,, \quad (85)$$

$$t_2 = \left[-x^{m+s}\phi_{m+s}\phi_t - \phi_{m+s}\phi_s\right]\left(\sigma^1\right)^2\,, \quad z_2^r = 0\,, \qquad z_2^{m+r} = \phi_r\phi_{m+r}\left(\sigma^1\right)^2\,. \qquad (86)$$

Here, the only constraint left at $\lambda = 0$ is $\phi_t = 0$.

## 5 Conclusions

In the present work, we have constructed the classical boundary field theory associated to the Jacobi sigma model by applying the poissonization procedure which gives a correspondence between the sought for field theory and the boundary field theory of the Poisson sigma model [11]. The next step is to quantize the field theory obtained here to obtain the holographic dual of the Jacobi sigma model. Since the discussion presented here is at the classical level, it is natural to ask what is the covariant quantization of the Jacobi sigma model and whether there is a relation between the quantum Jacobi and Poisson sigma theories. This question is not trivial since the quantization of the two theories imply the quantization of two different algebraic structures. We hope to report on that soon [21]. Other interesting problems related to the Jacobi sigma models are the existence of a AKSZ derivation in the sense of [22] and a thorough analysis of the existence of the induced morphism between the Jacobi algebroids in the sense of [19].

## Acknowledgments

I would like to thank to Shingo Takeuchi for very instructive discussions and to Thomas Strobl and Noriaki Ikeda for useful correspondence.

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
