# Peer review of "Classical boundary field theory of Jacobi sigma models by Poissonization"

_SciPost Physics_

## Round 2 · Referee Report · Anonymous (Referee 1) · 2021-2-14

Weaknesses
No new results.
Report
In this paper, the author constructs the classical boundary field theory of Jacobi sigma model, motivated by the possible applications in understanding the low-dimensional holographic principle. The construction is based on the poissonization procedure, i.e., on the interpretation of the Jacobi structure of a manifold M as a Poisson structure on M x R.
After short introduction, author reviews the Jacobi structure and the poissonization procedure in Section 2. In Section 3, the recent results on the construction of the Jacobi sigma model obtained in Refs.[2,1] are presented. As a main result, author derives the classical boundary field theory of Jacobi sigma model in Section 4. The classical fields of the model are expressed as perturbative expansion in terms of reduced phase space variables, closely following the corresponding construction for the Poisson sigma model of Ref.[16].
The analysis of the classical boundary field theory of the Jacobi sigma model presented in this paper follows trivially from the corresponding results for the Poisson sigma model. Essentially, the paper does not contain any new results, but could be accepted as a proceedings contribution after some revisions.
After short introduction, author reviews the Jacobi structure and the poissonization procedure in Section 2. In Section 3, the recent results on the construction of the Jacobi sigma model obtained in Refs.[2,1] are presented. As a main result, author derives the classical boundary field theory of Jacobi sigma model in Section 4. The classical fields of the model are expressed as perturbative expansion in terms of reduced phase space variables, closely following the corresponding construction for the Poisson sigma model of Ref.[16].
The analysis of the classical boundary field theory of the Jacobi sigma model presented in this paper follows trivially from the corresponding results for the Poisson sigma model. Essentially, the paper does not contain any new results, but could be accepted as a proceedings contribution after some revisions.
Requested changes
- List references in order in which they appear in text.
- Remove repeated text and equations. In Section 2, the sentence “The manifold M with the Jacobi structure is called a Jacobi manifold” appears twice on page 2. In Section 4, equations (56), (57-60) and (61-64) are essentially the same as (39), (40-43) and (44-47) from the previous section. The action (56) with the boundary condition (66) is the same as action (39). Therefore, equations of motions (57-60) following from the action (56) are the same as (40-43), and the gauge transformations (61-64) are identical to (44-47), up to the sign of gauge parameter. I suggest to keep the action in Section 4, but for eoms and gauge transformation to refer to Section 3.
- Correct equations and definitions. In Section 2, in the definition of the Schouten-Nijenhuis bracket, the result is p+q-1 vector, and not p+q as written. In eq.(9), the action of v on R is not defined. In text after eq.(13), 1-form gamma is called symplectic?! In section 3, in eqs.(31-34) and (36,37) a symbol for a map is used for a function depending on local coordinates, should be improved. At the end of Section 4, before eqs.(92-94), it is wrongly stated that “at lambda=0,1,2” – it should be of order 0,1,2 in lambda.
- Check the paper for typos. In equation (9), E should be exchanged with R, in Definition 4 p should be exchanged with rho, in eq. (25) beta on the rhs should be exchanged by gamma, in Definition 5, one letter is missing in item 2, word target.
- Specify in Introduction that the analysis in Section 4. follows closely the analysis of Ref.[16] for Poisson sigma model.

Author: Ion Vancea on 2021-02-17 [id 1250]
(in reply to Report 1 on 2021-02-14)Firstly, I would like to thank to the referee for carefully reading the manuscript and making objective and constructive suggestions.
In the current version of the manuscript, I have made all changes requested by the referee: 1. The references have been updated and arranged in the order of appearance. 2. The repeated text and formulas issues have been addressed as suggested by referee. 3. Corrections in the suggested formulas were done and the wrong formulations were corrected. Concerning the map notation in the equations of motion of the Poisson sigma model, I have introduce instead of it the coordinate notation and corrected the lambda statement. 4. I have corrected the typos. 5. I have added to the introduction the sentence "Our discussion follows closely the analysis from \cite{Vassilevich:2013ai} of the Poisson sigma model." which is the last but one sentence.
Also, I have removed uncited references and fixed some other minor typos.

---

## Round 4 · Referee Report · Anonymous · 2021-3-31

Report

The author revised the contribution according to the suggestions, therefore I find the contribution to the proceedings issue acceptable.

Requested changes

In the mean time, three references in the contribution, [1], [2] and [20], have been publish. Author might want to update the reference list.

---

## Editorial Decision

publication_decision_taken:_accept